# Type 2 Diabetes Prevention Focused on Normalization of Glycemia: A Two-Year Pilot Study

**DOI:** 10.3390/nu13030749

**Published:** 2021-02-26

**Authors:** Amy L McKenzie, Shaminie J Athinarayanan, Jackson J McCue, Rebecca N Adams, Monica Keyes, James P McCarter, Jeff S Volek, Stephen D Phinney, Sarah J Hallberg

**Affiliations:** 1Virta Health, San Francisco, CA 94105, USA; shaminie@virtahealth.com (S.J.A.); rebecca@virtahealth.com (R.N.A.); volek.1@osu.edu (J.S.V.); steve@virtahealth.com (S.D.P.); sarah@virtahealth.com (S.J.H.); 2University of Washington School of Medicine Wyoming WWAMI, Laramie, WY 82071, USA; jmmcue3@gmail.com; 3Department of Bariatric and Medical Weight Loss, Indiana University Health-Arnett, Lafayette, IN 47905, USA; keyesm@iuhealth.org; 4Department of Genetics, Washington University School of Medicine, St. Louis, MO 63110, USA; jamespmccarter@gmail.com; 5Abbott Diabetes Care, Inc., Alameda, CA 94502, USA; 6Department of Human Sciences, The Ohio State University, Columbus, OH 43210, USA

**Keywords:** prediabetes, remote continuous care, low carbohydrate, metabolic syndrome, obesity

## Abstract

The purpose of this study is to assess the effects of an alternative approach to type 2 diabetes prevention. Ninety-six patients with prediabetes (age 52 (10) years; 80% female; BMI 39.2 (7.1) kg/m^2^) received a continuous remote care intervention focused on reducing hyperglycemia through carbohydrate restricted nutrition therapy for two years in a single arm, prospective, longitudinal pilot study. Two-year retention was 75% (72 of 96 participants). Fifty-one percent of participants (49 of 96) met carbohydrate restriction goals as assessed by blood beta-hydroxybutyrate concentrations for more than one-third of reported measurements. Estimated cumulative incidence of normoglycemia (HbA1c < 5.7% without medication) and type 2 diabetes (HbA1c ≥ 6.5% or <6.5% with medication other than metformin) at two years were 52.3% and 3%, respectively. Prevalence of metabolic syndrome, class II or greater obesity, and suspected hepatic steatosis significantly decreased at two years. These results demonstrate the potential utility of an alternate approach to type 2 diabetes prevention, carbohydrate restricted nutrition therapy delivered through a continuous remote care model, for normalization of glycemia and improvement in related comorbidities.

## 1. Introduction

The United States faces a significant public health challenge with one in three adults living with prediabetes [1], a population at increased risk for progression to type 2 diabetes [2]. Patients with prediabetes often live with obesity and metabolic syndrome (MetS), each an independent predictor of type 2 diabetes [3,4], and the number of comorbidities is associated with increased risk of type 2 diabetes [5]. Each of these chronic conditions is associated with increased risk of cardiovascular disease, and evidence suggests microvascular damage may be present in patients with prediabetes prior to the development of obvious macrovascular disease. This demonstrates the need to initiate treatment for this high-risk state aimed at reversal of the condition to healthy or lower risk state to prevent or delay the onset of type 2 diabetes.

Intensive lifestyle intervention in the landmark Diabetes Prevention Program (DPP) reduced the incidence of type 2 diabetes by 58% [6], and use of behavioral interventions like the DPP are recommended by the United States Preventive Services Task Force to reduce risk [7]. Following the successful translation of the DPP into a community setting [8], the Centers for Disease Control (CDC) established the National Diabetes Prevention Program (NDPP) to make low-cost lifestyle interventions widely available, and the Centers for Medicare and Medicaid Services (CMS) determined that the NDPP met criteria for expansion to and reimbursement for Medicare participants [9]. For full CDC recognition and CMS reimbursement, NDPPs must meet specific operational criteria, including 5% average weight loss among participants enrolled at least nine months [10]. However, retention in these programs is severely challenged. The recent study by Cannon et al. of the NDPP observed only 31.9% retention at 10 months concurrent with a strong association between retention and weight loss [11]. These findings highlight the imminent need to reconsider the diabetes prevention strategy to ensure that meaningful health improvements are achieved more broadly across this high-risk population [12].

We developed an outcomes-driven program, focused on reducing hyperglycemia and normalization of glycemia to delay or prevent the progression to type 2 diabetes, rather than the 5% weight loss goal utilized in the NDPP. This intervention utilized carbohydrate-restricted nutrition therapy delivered through a remotely delivered continuous care model. In this pilot study among 96 patients with prediabetes, we aimed to assess the impact of this alternate approach to type 2 diabetes prevention on retention, adherence, and change in the metabolic condition status of prediabetes and related comorbidities over two years.

## 2. Materials and Methods

### 2.1. Design and Participants

Adults with medical record diagnoses of prediabetes and metabolic syndrome (*n* = 116) were enrolled in a single-arm, prospective, longitudinal study to assess the effects of the continuous care intervention on markers of metabolic health (Clinicaltrials.gov (accessed on 19 February 2021) Identifier NCT02519309). For the purpose of this analysis, prediabetes was defined as HbA1c < 6.5% concurrent with metformin use or HbA1c between 5.7% and 6.4%, inclusive, without the use of glycemic control medication to align with the American Diabetes Association Standards of Medical Care, given that metformin is recommended in patients with prediabetes [13]. Participants whose characteristics did not meet the defined criteria for prediabetes at baseline testing (*n* = 20) were excluded from the following analyses; this included patients whose baseline HbA1c was <5.7% without medication and patients who were found to be taking an antihyperglycemic medication other than metformin during the baseline history and physical assessment (Appendix A). Ninety-six participants were included in the analysis.

Participants between the ages of 21 and 65 years were recruited via clinical referrals, local media advertising, and word of mouth in Lafayette, Indiana and the surrounding area between August 2015 and March 2016. Individuals with advanced renal, hepatic, or cardiac dysfunction, dietary fat intolerance, or who were pregnant or planned to become pregnant were excluded from the study. The Franciscan Health Lafayette Institutional Review Board approved this study. All participants provided written informed consent.

### 2.2. Intervention

Details pertaining to the continuous care intervention were previously published [14,15,16]. In brief, participants accessed a mobile web-based application (app) which connected them to their remote care team consisting of a health coach who provided support for nutrition and behavior change and a medical provider who monitored the biomarkers and managed diabetes and hypertension medications. Participants self-selected to receive their education via either regularly scheduled on-site group classes consisting of presentations and group discussions or via web-based education modules consisting of videos and written materials viewed online at the participant’s choice of time and pace. The app also provided educational resources and access to peer social support via an online community regardless of the education delivery modality selected. Initial nutrition guidance included restricting dietary carbohydrates to fewer than 30 g per day, consumption of 1.5 g dietary protein per kg reference body weight daily, and consumption of dietary fat to satiety with the goal of achieving nutritional ketosis defined as blood beta-hydroxybutyrate (BHB) ≥ 0.5 mmol/L. The majority of dietary carbohydrates consisted of non-starchy vegetables, dairy, and/or nuts; participants selected individual foods based on their dietary preferences and philosophies. To monitor adherence to carbohydrate restriction and allow providers to manage medications, participants recorded blood glucose and BHB (Precision Xtra, Abbott; Alameda, CA, USA) and blood pressure (BP742 N, Omron Healthcare, Inc.; Lake Forest, IL, USA), if hypertension was diagnosed, in the app. Body weight was recorded in the app via cellular-connected scale (BT003, Body Trace; New York, NY, USA). Initially, participants measured and recorded biomarkers daily, and the care team adjusted the BHB target and frequency of reporting over time to meet individual health needs and goals.

### 2.3. Assessments

Participants underwent a history and physical examination and laboratory testing to obtain baseline and one- and two-year follow-up measures. Trained clinic staff assessed height, waist circumference, and blood pressure. Weight was uploaded to the app via a cellular connected scale provided to each participant. Trained staff at a Clinical Laboratory Improvement Amendment (CLIA) certified laboratory obtained blood from participants in a fasting state and analyzed blood samples for glucose, insulin, HDL-cholesterol (HDL-C), triglycerides, alanine transaminase (ALT), and aspartate aminotransferase (AST) on the day of sample collection or from stored serum.

We assigned the presence of conditions as follows: normoglycemia: HbA1c < 5.7% without glycemic control medication; prediabetes: HbA1c < 6.5% concurrent with metformin use or HbA1c between 5.7% and 6.4%, inclusive; type 2 diabetes: HbA1c ≥ 6.5% with or without glycemic control medication or HbA1c < 6.5% with glycemic control medication other than metformin; MetS: presence of three of five diagnostic criteria (BMI > 30 kg/m^2^ was substituted for waist circumference when it was not available) [17,18]; obesity ≥ class II: BMI ≥ 35 kg/m^2^; suspected hepatic steatosis: NAFLD-Liver Fat Score > −0.640 [19].

### 2.4. Statistical Methods

In this pilot study, we assessed the retention in the intervention and adherence to nutrition guidance. We assessed the outcome variables for assumptions of normality and linearity using Kline’s guidelines [20] and transformed variables as noted in the tables. We performed independent sample t-tests to examine the differences in baseline characteristics between those who selected on-site versus web-based education and between completers versus dropouts.

We calculated crude incidence of first occurrence of type 2 diabetes diagnosis and normoglycemia per 100 person-years and used the Kaplan–Meier approach to estimate the cumulative incidence [21] of type 2 diabetes and normoglycemia at two years. We assessed the changes in dichotomous outcome variables over time using generalized estimating equations (GEE) with binary logistic models and unstructured covariance matrices, and we estimated the missing values with 40 imputations [22] from logistic regression to allow intent-to-treat analysis. For continuous outcome variables, we utilized linear mixed effects models (LMM) to obtain the estimated marginal means and assess changes over the two-year follow-up period. The LMM uses an intent-to-treat principle which includes all available data and estimates the model parameters through a maximum-likelihood approach. An unstructured covariance matrix was specified. Covariates in GEE and LMM included baseline age, sex, race, and metformin use. LMM and chi-square were also utilized to assess the two-year clinical biomarker and retention differences, respectively, between those who selected on-site and web-based education. Significance level was set at 0.05 and was adjusted in each analysis with related variables to account for the number of contrasts using the Bonferroni method. We performed statistical analyses with SPSS statistical software (version 25.0, Armonk, NY, USA). Means are reported with (standard deviation) or ±standard error.

## 3. Results

### 3.1. Participant Characteristics, Retention, and Adherence

Participants with prediabetes were 52(10) years of age with a BMI of 39.24(7.06) kg/m^2^ at enrollment. Most participants were female (80%) and white/Caucasian (96%); four percent were African-American. Clinical characteristics among those who selected on-site versus web-based education were not different at baseline or two years (*p* > 0.05, Appendix A), nor was two-year retention (77.8% on-site vs. 71.4% web-based, *X*^2^ (1, *n* = 96) = 0.508, *p* = 0.476), so subsequent analyses were performed on the combined cohort. Metformin was prescribed to 15, 13, and 15 participants at baseline, one year, and two years, respectively, and thus was included as a covariate in statistical analyses.

Eighty percent of participants (77 of 96) remained enrolled in the intervention at one year, and 75% (72 of 96) at two years. Baseline clinical characteristics of two-year completers and dropouts were not different (Appendix A). Fifty-one percent of participants (49 of 96) obtained BHB ≥ 0.5 mmol/L for more than one-third of their reported measurements. Participants reported 205 ± 160 BHB measurements over two years.

### 3.2. Incidence of Normoglycemia and Type 2 Diabetes

Estimated cumulative incidence of normoglycemia at two years was 52.3%. The crude incidence for first occurrence of reversion from prediabetes to normoglycemia was 47.6 cases per 100 person-years. One new case of type 2 diabetes each year was observed in the population under study, resulting in a crude incidence of type 2 diabetes diagnosis of 1.5 cases per 100 person-years. The estimated cumulative incidence of type 2 diabetes at two years was 3%.

### 3.3. Change in Metabolic Condition Status

Prevalence of normoglycemia significantly increased, while prevalence of prediabetes, MetS, and suspected hepatic steatosis significantly decreased at one and two years (Table 1). The proportion of participants with class II and III obesity also significantly decreased (Figure 1). Prevalence of type 2 diabetes was unchanged from baseline after correction for multiple comparisons.

### 3.4. Change in Clinical Markers Associated with Metabolic Conditions

Clinical markers related to diabetes, obesity, and MetS improved except for blood pressure, in which a significant improvement was observed only in systolic pressure following one year (Table 2). At one and two years, 64% and 53% of participants enrolled, respectively, lost at least 5% body weight, and 54% and 47% lost at least 7%. Components of the NAFLD-Liver Fat Score (fasting insulin, aspartate aminotransferase, and alanine aminotransferase) for suspected steatosis significantly improved at one and two years except for aspartate aminotransferase, which was statistically unchanged.

## 4. Discussion

These results demonstrate the potential utility of an alternate approach to type 2 diabetes prevention, carbohydrate restricted nutrition therapy delivered through a continuous remote care model, for reversion of prediabetes and improvement of related comorbidities. Seventy-five percent of participants were retained in the program for two years, with an estimated cumulative incidence of normoglycemia of 52% and of progression to type 2 diabetes of 3%. Prevalence of MetS, class II and III obesity, and suspected hepatic steatosis within this cohort significantly declined. 

Retention in the present investigation was 80% and 75% at one and two years, respectively, far exceeding the 32% at 10 months [11] and 13.2% at one year [23] published in two different analyses of the NDPP. A number of factors may contribute to the differences observed. A remote delivery method may facilitate higher retention, as observed in another virtually delivered intervention [24]. Other factors include continuous access to a remote care team for support, daily focus on blood BHB goals rather than weight, and the magnitude of mean weight loss (12.7%) achieved in the first year. A relationship between weight loss and retention has been observed in both the NDPP and commercial weight loss programs [11,23,25]. Greater weight loss in the first year was associated with long-term weight loss maintenance of 5% or more, regardless of initial treatment, throughout the DPP and DPPOS [26].

Among participants in the present intervention, 64% and 53% achieved the ≥5% weight loss goal established by the CDC at one and two years, respectively, exceeding the 36% observed in the NDPP [23]. Nearly half of participants in the present study maintained ≥7% weight loss at two years, similar to the 24-week findings of the DPP, which declined to 38% at an average of 2.8 years follow-up [6]. Given the tendency for weight regain commonly observed across weight loss interventions, long-term retention and greater early weight loss in programs may play a critical role in helping participants maintain improved health status.

Achieving the 5% weight loss goal through a low fat, low calorie diet and physical activity goals has been the cornerstone of the NDPP given the relationship between weight loss and reduced risk of progression to type 2 diabetes in the DPP [27]. However, transient regression to normoglycemia in the first three years of the DPP was associated with significantly lower risk of progressing to type 2 diabetes during the 6–7 years of follow-up during the DPP Outcomes Study (DPPOS) [28]. The estimated cumulative incidence of reversion to normoglycemia (52%) in this study exceeded the approximately 35% observed at two years with intensive lifestyle intervention in the DPP [28]. Relatedly, incidence of progression to type 2 diabetes was low at 1.5 cases per 100 person-years, relative to 4.8 and 7.8 cases per 100-person years observed in the DPP lifestyle intervention and metformin groups [6]. These findings indicate that alternative short-term targets focused on normalization of glycemia, such as through dietary carbohydrate restriction, may provide viable alternatives to short-term diet and physical activity targets and longer-term weight loss (and weight loss maintenance) goals for diabetes prevention.

Reversion to normoglycemia is associated with positive health benefits beyond type 2 diabetes prevention or delay. Risk of cardiovascular disease, myocardial infarction, stroke, and all-cause mortality was reduced in a Chinese cohort of patients with prediabetes who reverted to normoglycemia within two years compared to those who progressed to type 2 diabetes over nearly nine years of follow-up [29]. In the DPPOS, achieving transient regression to normoglycemia also reduced odds of developing aggregate microvascular disease (retinopathy, nephropathy, and neuropathy), as well as retinopathy and nephropathy individually [30]. Prevalence of microvascular complications among the three DPP groups (lifestyle, metformin, and placebo) was similar at 15-years post-randomization as mean HbA1c across the groups converged to within 0.3% and above 6.0%, but prevalence of microvascular complications was 28% lower among those who did not progress to type 2 diabetes compared to those who did [31]. This may suggest a key role for long-term maintenance of normoglycemia or prevention of progression to type 2 diabetes for maximum benefit. Considering the high rates of retention and normalization of glycemia observed in this study combined with the remote delivery and monitoring methods utilized, this intervention may have the potential to address a critical need in this high-risk population, and future research should assess its long-term effects on prevention of type 2 diabetes and its complications.

Although meeting a particular weight loss target was not a stated goal for participants in this intervention, the majority of enrolled participants met the 5% benchmark at two years. Lifestyle intervention independent of weight loss predicted regression to normoglycemia in the DPP [32], and hyperglycemia can be resolved prior to significant weight loss following bariatric surgery [33]. Further, carbohydrate restriction in the absence of weight loss has been demonstrated to reverse metabolic syndrome [34]. Taken together, this may suggest that weight loss can be an effect of metabolic health improved by other means, rather than a primary driver, further highlighting the potential for alternate goals related to the ultimate outcome of diabetes prevention. 

Accompanying normalization of glycemia and weight loss, prevalence of MetS and suspected hepatic steatosis declined following this intervention. Reduction in the prevalence of MetS (−45%) exceeded that of the DPP, where prevalence declined from 51 to 43% [35] and was similar to a four-week low-carbohydrate feeding study [34], which demonstrated that MetS resolution is possible with carbohydrate restriction even in the absence of weight loss. Similarly, a study in patients with NAFLD demonstrated that liver fat was reduced significantly following just one day of consuming a ketogenic diet due to reduced de novo lipogenesis and increased beta oxidation [36], providing a potential explanation for the decreased prevalence of suspected hepatic steatosis observed in this study. The inverse trend in some biomarkers between one and two years is of unknown significance given the significant improvement maintained at two years compared to baseline and existing evidence demonstrating that even transient normalization of glucose can have long-term positive health benefit.

Strengths of this study include its two-year follow-up period and assessment of incident type 2 diabetes, which is lacking in the NDPP. Limitations include the predominance of females enrolled in the study (although this is similar to enrollment in the NDPP), the lack of racial diversity, and that the study was not designed to test the contribution of each component of the intervention to outcomes, nor to evaluate equivalence or superiority to alternate interventions or care models. Data were analyzed conservatively according to intent-to-treat principles and included participants who did not fully adhere to the intervention components; thus, these outcomes are likely to reflect what might be expected in a real-world setting.

As observed in the DPP, clinical outcomes are often tied to program retention and adherence, but focus should remain on achieving and sustaining clinically meaningful outcomes. Historically in the context of prediabetes, outcomes have focused on a 5% weight loss goal through adhering to a low fat, low calorie diet and physical activity targets, but evidence now demonstrates that metabolic health can be improved by focusing on alternate targets, such as achievement of normoglycemia through nutrition therapy. Remote delivery methods may provide another strategy for improving retention and facilitating improved health outcomes in a larger proportion of individuals. 

## 5. Conclusions

This pilot study demonstrated that the majority of patients with prediabetes who chose to enroll in this intervention achieved normoglycemia and maintained clinically meaningful weight loss through two years, suggesting this intervention utilizing carbohydrate restricted nutrition therapy delivered through a continuous remote care model may provide an additional and alternative approach for type 2 diabetes prevention. Future research may evaluate the effectiveness of this care model versus alternatives for the prevention or delay of progression to type 2 diabetes.

## Figures and Tables

**Figure 1 nutrients-13-00749-f001:**
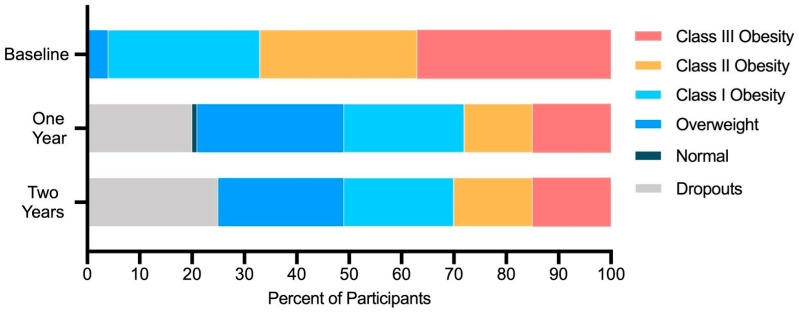
Prevalence of obesity classes and body mass index categories among participants over time.

**Table 1 nutrients-13-00749-t001:** Prevalence of metabolic condition status over two years.

Metabolic Condition	Baseline			1 Year			2 Years	
	*n*	Mean ± SE	*n*	Mean ± SE	*p*	*n*	Mean ± SE	*p*
Prediabetes (%)	96	100.0 ± 0.0	70	54.0 ± 6.0	<0.001	63	67.0 ± 5.9	<0.001
Normoglycemia (%)	96	0.0 ± 0.0	70	46.0 ± 6.0	<0.001	63	33.0 ± 5.9	<0.001
Type 2 Diabetes (%)	96	0.0 ± 0.0	70	4.0 ± 2.7	0.04	63	5.0 ± 3.1	0.02
Metabolic Syndrome (%)	94	94.0 ± 2.5	65	30.0 ± 5.7	<0.001	47	49.0 ± 7.1	<0.001
Obesity ≥ Class II (%)	96	67.0 ± 4.8	77	38.0 ± 5.5	<0.001	72	43.0 ± 5.6	<0.001
Suspected Steatosis (%)	89	88.0 ± 3.5	58	41.0 ± 6.1	<0.001	42	48.0 ± 6.5	<0.001

Note: *n* indicates the available data at the time point. Multiple imputation was utilized to facilitate intent-to-treat analysis. Contrasts compared follow-up to baseline. Statistical significance is indicated by *p* < 0.004 following Bonferroni correction for multiple comparisons.

**Table 2 nutrients-13-00749-t002:** Change in metabolic condition clinical markers compared to baseline.

		Baseline	1 year			2 years		
	*n*	EMM ± SE	*n*	EMM ± SE	*p*	*n*	EMM ± SE	*p*
HbA1c (%)	96	5.95 ± 0.02	70	5.63 ± 0.03	<0.001	64	5.73 ± 0.04	<0.001
HbA1c (mmol/mol)	96	41.5 ± 0.2	70	38.3 ± 0.3	<0.001	64	39.3 ± 0.4	<0.001
Fasting Glucose (mmol/L)	95	6.11 ± 0.08	69	5.61 ± 0.08	<0.001	63	5.64 ± 0.08	<0.001
Fasting Insulin (pmol/L)	90	164.80 ± 10.21	67	94.73 ± 6.53	<0.001	58	104.59 ± 7.22	<0.001
SBP (mmHg)	95	129.9 ± 1.4	62	123.1 ± 1.5	<0.001	48	127.3 ± 1.8	0.18
DBP (mmHg)	95	82.5 ± 0.8	62	79.2 ± 1.0	0.01	48	80.5 ± 1.1	0.11
Weight (kg)	96	109.6 ± 2.2	77	95.7 ± 1.9	<0.001	72	97.2 ± 1.9	<0.001
BMI (kg/m^2^)	96	39.08 ± 0.72	77	34.11 ± 0.63	<0.001	72	34.62 ± 0.62	<0.001
Waist Circumference (cm)	74	118.9 ± 1.6	52	107.8 ± 1.7	<0.001	42	110.9 ± 2.7	0.002
HDL-cholesterol (mmol/L)	90	1.28 ± 0.03	67	1.45 ± 0.04	<0.001	58	1.46 ± 0.05	<0.001
Triglycerides (mmol/L)	90	1.81 ± 0.09	67	1.38 ± 0.09	<0.001	58	1.28 ± 0.08	<0.001
ALT (µkat/L) †	95	0.46 ± 0.02	69	0.37 ± 0.02	<0.001	63	0.37 ± 0.02	<0.001
AST (µkat/L) †	95	0.37 ± 0.02	69	0.34 ± 0.02	0.03	63	0.33 ± 0.01	0.04
NAFLD-Liver Fat Score	89	1.84 ± 0.24	58	−0.78 ± 0.20	<0.001	42	−0.35 ± 0.24	<0.001

Note: *n* indicates the available data at the time point. Contrasts compared follow-up to baseline. Statistical significance is indicated by *p* < 0.002 following Bonferroni correction for multiple comparisons. Abbreviations: SBP, systolic blood pressure; DBP, diastolic blood pressure; HDL, high density lipoprotein; ALT, alanine aminotransferase; AST, aspartate aminotransferase, NAFLD, non-alcoholic fatty liver disease. † Variable failed normality (positively skewed). Analyses were conducted on data excluding the top 1% of values and treating these values as missing in the LMM model.

## Data Availability

Data are not publicly available due to privacy concerns.

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
