# Peer review of "Type 2 Diabetes Prevention Focused on Normalization of Glycemia: A Two-Year Pilot Study"

_nutrients, 2021, doi:10.3390/nu13030749_

Round 1

Reviewer 1 Report

The authors have evaluated an alternative approach in the treatment of type 2 diabetes mellitus. The title indicates "Prevention of diabetes mellitus 2" although after reading the work it seems that it is more about treatment than prevention. There are also methodological aspects that should be explained.

TITLE:

Type 2 diabetes prevention focused on normalization of glycemia: a two-year pilot study.

The work is focused on treating people with type 2 diabetes or prediabetes rather than preventing the disease.

  1. Materials and Methods

Design and Participants

The authors consider that persons with prediabetes are patients with an HbA1c concentration < 6.5 % whether they receive treatment with metformin or not, and these are totally different clinical situations. The ADA considers prediabetes an HbA1c value between 5.7 and 6.4%

Initial nutrition guidance included restricting dietary carbohydrates to fewer than 30 grams per day, consumption of 1.5 g dietary protein per kg reference body weight daily, and consumption of dietary fat to satiety with the goal of achieving nutritional ketosis defined as blood beta-hydroxybutyrate (BHB) ≥ 0.5 mmol/L.

Authors should explain the carbohydrate foods that were included in the diet and the amounts.

Among the objectives of the dietary recommendations is to re-educate dietary habits and acquire habits that can be maintained over time. The authors consider that this type of diet is advisable in the medium or long term?.

The daily amount of carbohydrates is very low and the risk of complications from ketosis is very high. The authors should explain the follow-up protocol for possible complications from ketosis.

Weight was uploaded to the app via a cellular connected scale provided to each participant.

It appears that the patients were not weighed during the physical examination. Why?

Trained staff at a Clinical Laboratory Improvement Amendment (CLIA) certified laboratory obtained blood from participants in a fasting state and completed analyses of blood samples on the day of sample collection or from stored serum.

The laboratory measurements are missing: Glucose, insulin, HDL-cholesterol, tryglycerides, ALT, and AST.

Why LDL-cholesterol was not measured. A fat diet was prescribed to satiety. Measuring the concentration of total cholesterol or LDL-cholesterol would be necessary since a diet rich in fat is followed.

We assigned the presence of conditions as follows: normoglycemia: HbA1c <5.7%  without glycemic control medication; prediabetes: HbA1c<6.5% concurrent with metformin use or HbA1c between 5.7% and 6.4%, inclusive; type 2 diabetes: HbA1c ≥ 6.5% with or without glycemic control medication or HbA1c < 6.5% with glycemic control medication other than metformin;

Metformin is a treatment for diabetes mellitus 2.

MetS: presence of three of five diagnostic criteria (BMI > 30 kg/m2 was substituted for waist circumference when it was not available)

Why was waist circumference unavailable? Waist circumference was included in the study protocol.

The study was really prospective or used a database and therefore was a retrospective design.

  1. Results

How many patients were on drug treatment throughout the study.

Most participants (92%; 88/96) reported achieving BHB ≥0.5 mmol/L at least once during the two years,

At least once in 2 years is irrelevant

and 51% (49/96) of participants obtained BHB ≥0.5 mmol/L for more than one-third of their reported measurements.

This figure does not indicate strict adherence to the nutritional guide.

It is not stated how many reports there were on average per patient.

Table 1.

Prevalence measures: in rows indicates that the figures are expressed in % and in columns Mean ± SE. ¿?

What groups are compared?

The Kaplan-Meier curves that are indicated in statistical methods are not shown in the results section.

  1. Discussion

The results do not show how many patients received pharmacological treatment and the dose of drugs, therefore the result cannot be attributed only to the effect of the nutritional intervention.

Author Response

Reviewer 1

The authors have evaluated an alternative approach in the treatment of type 2 diabetes mellitus. The title indicates "Prevention of diabetes mellitus 2" although after reading the work it seems that it is more about treatment than prevention. There are also methodological aspects that should be explained.

Thank you for this feedback and efforts to improve the manuscript.  Please find a point-by-point response below.

Please note a new publication in J Am Heart Assoc (26 Jan 2021) relevant to the discussion was added in this revised version of the manuscript (~lines 239-243).

TITLE: Type 2 diabetes prevention focused on normalization of glycemia: a two-year pilot study.

The work is focused on treating people with type 2 diabetes or prediabetes rather than preventing the disease.

Thank you for sharing this perspective. People with type 2 diabetes were not included in this analysis. Historically, interventions focused on type 2 diabetes prevention have been studied in people at high risk for diabetes, often with elevated fasting glucose or elevated 2h plasma glucose following OGTT above normal but not high enough to warrant a type 2 diabetes diagnosis (for example, see inclusion criteria in the primary Diabetes Prevention Program publication Knowler WC et al. NEJM. 2002; 346:393–403). Although treatment of prediabetes may be an appropriate phrase since glucose and several other risk factors for diabetes normalized in many participants, we felt it was appropriate to utilize the conventional terminology of the field (i.e., type 2 diabetes prevention).

  1. Materials and Methods

Design and Participants

The authors consider that persons with prediabetes are patients with an HbA1c concentration < 6.5 % whether they receive treatment with metformin or not, and these are totally different clinical situations. The ADA considers prediabetes an HbA1c value between 5.7 and 6.4%

Thank you for bringing this to our attention. We’ve updated the text to better clarify this rationale (~line 67).

For the purpose of this analysis, prediabetes was defined as HbA1c <6.5% concurrent with metformin use or HbA1c between 5.7% and 6.4%, inclusive, without the use of glycemic control medication to align with the American Diabetes Association Standards of Medical Care, given that metformin is recommended in patients with prediabetes [13].

Initial nutrition guidance included restricting dietary carbohydrates to fewer than 30 grams per day, consumption of 1.5 g dietary protein per kg reference body weight daily, and consumption of dietary fat to satiety with the goal of achieving nutritional ketosis defined as blood beta- hydroxybutyrate (BHB) ≥ 0.5 mmol/L.

Authors should explain the carbohydrate foods that were included in the diet and the amounts.

Thank you, this has been clarified in the text (~line 94).

Initial nutrition guidance included restricting dietary carbohydrates to fewer than 30 grams per day, consumption of 1.5 g dietary protein per kg reference body weight daily, and consumption of dietary fat to satiety with the goal of achieving nutritional ketosis defined as blood beta-hydroxybutyrate (BHB) ≥ 0.5 mmol/L. The majority of dietary carbohydrates consisted of non-starchy vegetables, dairy, and/or nuts; participants selected individual foods based on their dietary preferences and philosophies. 

Among the objectives of the dietary recommendations is to re- educate dietary habits and acquire habits that can be maintained over time. The authors consider that this type of diet is advisable in the medium or long term?.

This comment isn’t clear to the authors. Please re-state.

The daily amount of carbohydrates is very low and the risk of complications from ketosis is very high. The authors should explain the follow-up protocol for possible complications from ketosis.

Very high ketone levels (often ≥15 mM) signal the pathological condition of ketoacidosis or metabolic acidosis. On the other hand, low to moderate physiological concentrations of beta-hydroxybutyrate (such as the 0.5-3.0 mM of endogenously produced ketones in this study - often referred to as “nutritional ketosis”) are a normal metabolic state. No complications associated with low to moderate levels of ketosis have been observed. Ketones serve as an alternate fuel source to glucose and growing evidence suggests beta-hydroxybutyrate has signalling properties that can reduce inflammation and improve insulin resistance (Newman JC & Verdin E. Ketone bodies as signaling metabolites. Trends Endocrinol Metabolism. 2014; 25:42–52.).

Weight was uploaded to the app via a cellular connected scale provided to each participant.

It appears that the patients were not weighed during the physical examination. Why?

Thank you for this comment. Weight was measured during physical examinations, but we chose to use weights automatically sent to the app from the participants’ home scale for analysis, as participants often weighed themselves at home with consistent methods. The weight is sent automatically via cellular connection and does not rely on self-report or recall. Further, the intra-class correlation (ICC) was high (>0.90) between the app-uploaded weight and clinic-measured weight at all time points confirming a high agreement between these two weight variables. 

Trained staff at a Clinical Laboratory Improvement Amendment (CLIA) certified laboratory obtained blood from participants in a fasting state and completed analyses of blood samples on the day of sample collection or from stored serum.

The laboratory measurements are missing: Glucose, insulin, HDL- cholesterol, triglycerides, ALT, and AST.

Thank you for this feedback. We revised the text to address this (~line 110).

Trained staff at a Clinical Laboratory Improvement Amendment (CLIA) certified laboratory obtained blood from participants in a fasting state and analyzed blood samples for glucose, insulin, HDL-cholesterol (HDL-C), triglycerides, alanine transaminase (ALT), and aspartate aminotransferase (AST) on the day of sample collection or from stored serum. 

Reviewer 2 Report

The work of McKenzie et al. aims to evaluate the impact of a nutritional therapy program with carbohydrate restriction, delivered continuously and remotely for two years, on the retention, adherence, and alteration of the metabolic status of patients with prediabetes and related comorbidities. For this purpose, the authors conducted a prospective, longitudinal, single-arm study. 96 individuals with pre-diabetes (PD) diagnosis (94% with Metabolic Syndrome (MS)) at baseline were submitted to a remote nutritional intervention supported by a care team and received educational modules either through on-site group classes or web-based educational modules.

This is a well-structured and written article on a relevant topic. Prevention of the progression of dysglycemia is crucial to reverse the current T2D landscape, and the authors have chosen an interesting and timely approach.

Major comments:

It is widely accepted that T2D, as well as PD, are heterogeneous conditions and it has been suggested that its subgroups should be approached differently. In this context, digital health and remotely delivered care can enable us to provide “precise” patient care. It would be important to critically review the results in this context, namely the need for a “precise” nutritional methodology.

The authors show a statistically significant decrease in the prevalence of PD, MS, Class of Obesity, and Suspicion of Steatosis from baseline to 1st and 2nd year of intervention. However, when comparing 1st and 2nd-year results, most of them seem to have an inverse trend. The authors should comment on this fact, especially due to the importance of sustained results, given that the risk of T2D complications also depends on cumulative exposure over time.

Although the authors acknowledge that the study was not intended to test the difference for alternative care models, this is also a limitation of the study. The authors suggest that the model of care used may be an alternative approach to others, however, some questions remain: how does it compare with other nutritional programs and other non-hospital care programs?

Minor Comments:

Line14 - "Fifty-one percent (49/96)". 49 should be corrected.

Line17 - The definition of type 2 diabetes includes <6.5%HbA1c with drugs other than metformin.

Line79 - What does "on-site group classes" mean? Presential or digital (video, for example)?

Reviewer 3 Report

General comments:

Overall, this is a well written, clearly presented and overall interesting short report that highlights the impact of an alternative approach to type 2 diabetes prevention. McKenzie AL. et al demonstrate the potential utility of a carbohydrate restricted nutrition therapy (delivered through a continuous remote care model) for glycemic control, metabolic syndrome and related comorbidities.

  1. Given the importance of metabolic syndrome, a little more information regarding the condition is warranted in the introduction. It is only introduced as a predictor of T2D. The general reader would benefit from more detail.
  2. Line 30. Missing the word “by”. Reduced incidence of type 2 diabetes 58%.
  3. Though not significant it is interesting that the “dropouts/data missing” individuals had higher waist circumference (p=0.083) and higher triglycerides (p=0.085) at baseline. These individuals would have clearly benefited from the intervention. Is there partial data for these individuals? It would be interesting to examine.

Round 2

Reviewer 1 Report

Thank you very much for correcting some aspects but there are still issues that need to be clarified

The participants could therefore consume dairy products, fruits, vegetables and nuts and they selected the foods according to their preferences. This does not ensure that they consumed 30 g / d of carbohydrates. Only with milk and fruit they already exceed 30 g.

MetS: presence of three of five diagnostic criteria (BMI > 30 kg/m2 was substituted for waist circumference when it was not available)

Why was waist circumference unavailable? Waist circumference was included in the study protocol.

The study was really prospective or used a database and therefore was a retrospective design.

  1. Results

How many patients were on drug treatment throughout the study.

Most participants (92%; 88/96) reported achieving BHB ≥0.5 mmol/L at least once during the two years,

At least once in 2 years is irrelevant

and 51% (49/96) of participants obtained BHB ≥0.5 mmol/L for more than one-third of their reported measurements.

This figure does not indicate strict adherence to the nutritional guide.

It is not stated how many reports there were on average per patient.

Table 1.

Prevalence measures: in rows indicates that the figures are expressed in % and in columns Mean ± SE. ¿?

What groups are compared?

The Kaplan-Meier curves that are indicated in statistical methods are not shown in the results section.

  1. Discussion

The results do not show how many patients received pharmacological treatment and the dose of drugs, therefore the result cannot be attributed only to the effect of the nutritional intervention.

Author Response

Thank you very much for correcting some aspects but there are still issues that need to be clarified

Thank you for your quick response. Our sincere apologies, it seems the latter half of our responses were cut off. Please find our point by point responses below to the new comment and the latter comments from the first round of review.

The participants could therefore consume dairy products, fruits, vegetables and nuts and they selected the foods according to their preferences. This does not ensure that they consumed 30 g / d of carbohydrates. Only with milk and fruit they already exceed 30 g.

There seems to be a misunderstanding. The text states:

"Initial nutrition guidance included restricting dietary carbohydrates to fewer than 30 grams per day, consumption of 1.5 g dietary protein per kg reference body weight daily, and consumption of dietary fat to satiety with the goal of achieving nutritional ketosis defined as blood beta-hydroxybutyrate (BHB) ≥ 0.5 mmol/L. The majority of dietary carbohydrates consisted of non-starchy vegetables, dairy, and/or nuts; participants selected individual foods based on their dietary preferences and philosophies." 

We agree that fruit and milk would make it difficult to aim for fewer than 30 grams of carbohydrates per day initially, and thus, these items were not recommended. Examples of dairy that participants may have selected to consume would be cheese, full fat yogurt, or heavy cream. 

MetS: presence of three of five diagnostic criteria (BMI > 30 kg/m2 was substituted for waist circumference when it was not available)

Why was waist circumference unavailable? Waist circumference was included in the study protocol.

Unfortunately, some waist circumference measurements were missed during data collection. When these data were missing, we utilized the approach recommended by the International Diabetes Federation (IDF) where central obesity is assumed if their BMI > 30kg/m2

The study was really prospective or used a database and therefore was a retrospective design.

This was a prospective study, as stated in the methods.

  1. Results

How many patients were on drug treatment throughout the study.

Approximately 16% of the participants were prescribed metformin at baseline, informing the decision to include metformin use as a covariate in the GEE and LMM analyses (as stated in the statistical methods ~line 138). Results regarding metformin prescription throughout the study were added to the results section (~line 153).

The revised text states:

"Metformin was prescribed to 15, 13, and 15 participants at baseline, one year, and two years, respectively, and thus was included as a covariate in statistical analyses."

Most participants (92%; 88/96) reported achieving BHB ≥0.5 mmol/L at least once during the two years,

At least once in 2 years is irrelevant

This has been removed.

and 51% (49/96) of participants obtained BHB ≥0.5 mmol/L for more than one-third of their reported measurements.

This figure does not indicate strict adherence to the nutritional guide.

It is not stated how many reports there were on average per patient.

Thank you for bringing this to our attention. We added descriptive statistics regarding the number of BHB measurements reported (~line 159). To the best of our knowledge, there is no interpretation or conclusion drawn in the manuscript that suggests strict adherence to this guide was observed. We did notice upon review that the methods regarding the BHB goal was unclear, and we clarified that while the BHB target was initially 0.5 mM, both the BHB target and reporting frequency were adjusted by the care team according to the individual health needs and goals of the patient (~line 102).

The revised text states:

"Fifty-one percent of participants (49 of 96) obtained BHB ≥0.5 mmol/L for more than one-third of their reported measurements. Participants reported 205±160 BHB measurements over two years."

"Initially, participants measured and recorded biomarkers daily, and the care team adjusted the BHB target and frequency of reporting over time to meet individual health needs and goals." 

Table 1.

Prevalence measures: in rows indicates that the figures are expressed in % and in columns Mean ± SE. ¿?

For the assessment of the metabolic condition prevalence as a binary variable at each time point, we used generalized estimating equations (GEE). This model can be used for both continuous and categorical variables. The proportion (or percentage as we report in this case) of individuals categorized in different metabolic conditions was estimated for each time point after accounting for covariates that were included in the model. Since the model provides a mean estimate of the proportion, it also includes a GEE estimate of a robust standard error which is calculated based on the actual variations in the cluster-level statistics. 

What groups are compared?

Thank you for bringing this to our attention. We assessed differences between the follow up time point (1 or 2 years) and baseline and noted this in the table legends. 

The Kaplan-Meier curves that are indicated in statistical methods are not shown in the results section.

The Kaplan-Meier approach was used to estimate the cumulative incidence of type 2 diabetes and normoglycemia at 2 years, which is stated in the results text (~line 161). 

  1. Discussion

The results do not show how many patients received pharmacological treatment and the dose of drugs, therefore the result cannot be attributed only to the effect of the nutritional intervention.

Thank you for bringing this to our attention. In this analysis, the prescription of any diabetes medication other than metformin would indicate progression to type 2 diabetes, as stated in the methods section. The progression to type 2 diabetes observed in this study was very low (one case each year). Metformin was prescribed to a small number of participants (16% of the sample at baseline), remained relatively stable over two years, and was accounted for as a covariate in statistical analyses. The proportion of participants prescribed metformin was added to the results section (~line 153). Given these factors, the interpretation of results appears appropriate.